# Learning from Between-class Examples for Deep Sound Recognition

**Yuji Tokozume**[1], **Yoshitaka Ushiku**[1], **Tatsuya Harada**[1,2]
[1]The University of Tokyo, [2]RIKEN
{tokozume,ushiku,harada}@mi.t.u-tokyo.ac.jp

## Abstract

Deep learning methods have achieved high performance in sound recognition tasks. Deciding how to feed the training data is important for further performance improvement. We propose a novel learning method for deep sound recognition: Between-Class learning (*BC learning*). Our strategy is to learn a discriminative feature space by recognizing the between-class sounds as between-class sounds. We generate between-class sounds by mixing two sounds belonging to different classes with a random ratio. We then input the mixed sound to the model and train the model to output the mixing ratio. The advantages of BC learning are not limited only to the increase in variation of the training data; BC learning leads to an enlargement of Fisher's criterion in the feature space and a regularization of the positional relationship among the feature distributions of the classes. The experimental results show that BC learning improves the performance on various sound recognition networks, datasets, and data augmentation schemes, in which BC learning proves to be always beneficial. Furthermore, we construct a new deep sound recognition network (*EnvNet-v2*) and train it with BC learning. As a result, we achieved a performance surpasses the human level[1].

## 1 Introduction

Sound recognition has been conventionally conducted by applying classifiers such as SVM to local features such as MFCC or log-mel features (Logan et al., 2000; Vacher et al., 2007; Łopatka et al., 2010). Convolutional neural networks (CNNs) (LeCun et al., 1998), which have achieved success in image recognition tasks (Krizhevsky et al., 2012; Simonyan & Zisserman, 2015; He et al., 2016), have recently proven to be effective in tasks related to series data, such as speech recognition (Abdel-Hamid et al., 2014; Sainath et al., 2015a;b) and natural language processing (Kim, 2014; Zhang et al., 2015). Some researchers applied CNNs to sound recognition tasks and achieved high performance (Aytar et al., 2016; Dai et al., 2017; Tokozume & Harada, 2017).

The amount and quality of training data and how to feed it are important for machine learning, particularly for deep learning. Various approaches have been proposed to improve the sound recognition performance. The first approach is to efficiently use limited training data with data augmentation. Researchers proposed increasing the training data variation by altering the shape or property of sounds or adding a background noise (Tokozume & Harada, 2017; Salamon & Bello, 2017). Researchers also proposed using additional training data created by mixing multiple training examples (Parascandolo et al., 2016; Takahashi et al., 2016). The second approach is to use external data or knowledge. Aytar et al. (2016) proposed learning rich sound representations using a large amount of unlabeled video datasets and pre-trained image recognition networks. The sound dataset expansion was also conducted (Salamon et al., 2014; Piczak, 2015b; Gemmeke et al., 2017).

In this paper, as a novel third approach we propose a learning method for deep sound recognition: Between-Class learning (*BC learning*). Our strategy is to learn a discriminative feature space by recognizing the between-class sounds as between-class sounds. We generate between-class sounds by mixing two sounds belonging to different classes with a random ratio. We then input the mixed sound to the model and train the network to output the mixing ratio. Our method focuses on the characteristic of the sound, from which we can generate a new sound simply by adding the waveform

---

[1]The code is publicly available at https://github.com/mil-tokyo/bc_learning_sound/.

data of two sounds. The advantages of BC learning are not limited only to the increase in variation of the training data; BC learning leads to an enlargement of Fisher's criterion (Fisher, 1936) (*i.e.*, the ratio of the between-class distance to the within-class variance) in the feature space, and a regularization of the positional relationship among the feature distributions of the classes.

The experimental results show that BC learning improves the performance on various sound recognition networks, datasets, and data augmentation schemes, in which BC learning proves to be always beneficial. Furthermore, we constructed a new deep sound recognition network (*EnvNet-v2*) and trained it with BC learning. As a result, we achieved a $15.1\%$ error rate on a benchmark dataset ESC-50 (Piczak, 2015b), which surpasses the human level.

We argue that our BC learning is different from the so-called data augmentation methods we introduced above. Although BC learning can be regarded as a data augmentation method from the viewpoint of using augmented data, the novelty or key point of our method is not mixing multiple sounds, but rather learning method of training the model to output the mixing ratio. This is a fundamentally different idea from previous data augmentation methods. In general, data augmentation methods aim to improve the generalization ability by generating additional training data which is likely to appear in testing phase. Thus, the problem to be solved is the same in both training and testing phase. On the other hand, BC learning uses only mixed data and labels for training, while mixed data does not appear in testing phase. BC learning is a method to improve the classification performance by solving a problem of predicting the mixing ratio between two different classes. To the best of our knowledge, this is the first time a learning method that employs a mixing ratio between different classes has been proposed. We intuitively describe why such a learning method is effective and demonstrate the effectiveness of BC learning through wide-ranging experiments.

## 2   RELATED WORK

### 2.1   SOUND RECOGNITION NETWORKS

We introduce recent deep learning methods for sound recognition. Piczak (2015a) proposed to apply CNNs to the log-mel features extracted from raw waveforms. The log-mel feature is calculated for each frame of sound and represents the magnitude of each frequency area, considering human auditory perception (Davis & Mermelstein, 1980). Piczak created a 2-D feature-map by arranging the log-mel features of each frame along the time axis and calculated the delta log-mel feature, which was the first temporal derivative of the static log-mel feature. Piczak then classified these static and delta feature-maps with 2-D CNN, treating them as a two-channel input in a manner quite similar to the RGB inputs of the image. The log-mel feature-map exhibits locality in both time and frequency domains (Abdel-Hamid et al., 2014). Therefore, we can accurately classify this feature-map with CNN. We refer to this method as *Logmel-CNN*.

Some researchers also proposed methods to learn the sounds directly from 1-D raw waveforms, including feature extraction. Aytar et al. (2016) proposed a sound recognition network using 1-D convolutional and pooling layers named SoundNet and learned the sound feature using a large amount of unlabeled videos (we describe the details of it in the next section). Dai et al. (2017) also proposed a network using 1-D convolutional and pooling layers, but they stacked more layers. They reported that the network with 18 layers performed the best. Tokozume & Harada (2017) proposed a network using both 1-D and 2-D convolutional and pooling layers named EnvNet. First, EnvNet extracts a frequency feature of each short duration of section with 1-D convolutional and pooling layers and obtain a 2-D feature-map. Next, it classifies this feature-map with 2-D convolutional and pooling layers in a similar manner to Logmel-CNN. Learning from the raw waveform is still a challenging problem because it is difficult to learn raw waveform features from limited training data. However, the performance of these systems is close to that of Logmel-CNN.

### 2.2   APPROACHES TO ACHIEVE HIGH PERFORMANCE

We describe the approaches to achieve high sound recognition performance from two views: approaches involving efficient use of limited training data and those involving external data/knowledge. First, we describe data augmentation as an approach of efficiently using limited training data. One of the most standard and important data augmentation methods is cropping (Piczak, 2015a; Aytar et al., 2016; Tokozume & Harada, 2017). The training data variation increases, and we are able to more

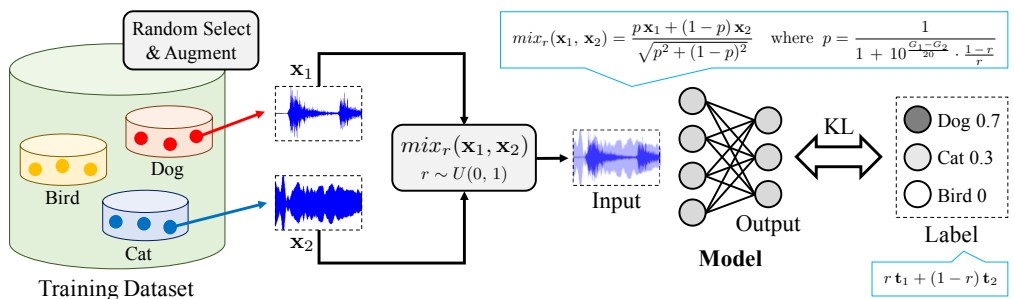

Figure 1: Pipeline of BC learning. We create each training example by mixing two sounds belonging to different classes with a random ratio. We input the mixed sound to the model and train the model to output the mixing ratio using the KL loss.

efficiently train the network when the short section (approximately 1–2 s) of the training sound cropped from the original data, and not the whole section, is input to the network. A similar method is generally used in the test phase. Multiple sections of test data are input with a stride, and the average of the output predictions is used to classify the test sound. Salamon & Bello (2017) proposed the usage of additional training data created by time stretching, pitch shifting, dynamic range compression, and adding background noise chosen from an external dataset. Researchers also proposed using additional training data created by mixing multiple training examples. Parascandolo et al. (2016) applied this method to polyphonic sound event detection. Takahashi et al. (2016) applied this method to single-label sound event classification, but only the sounds belonging to the same class were mixed. Our method is different from both of them in that we employ a mixing ratio between different classes for training.

Next, we describe the approaches of utilizing external data/knowledge. Aytar et al. (2016) proposed to learn rich sound representations using pairs of image and sound included in a large amount of unlabeled video dataset. They transferred the knowledge of pre-trained large-scale image recognition networks into sound recognition network by minimizing the KL-divergence between the output predictions of the image recognition networks and that of the sound network. They used the output of the hidden layer of the sound recognition network as the feature when applying to the target sound classification problem. They then classified it with linear SVM. They could train a deep sound recognition network (SoundNet8) and achieve a $74.2\%$ accuracy on a benchmark dataset, ESC-50 (Piczak, 2015b), with this method.

## 3 BETWEEN-CLASS LEARNING FOR SOUND RECOGNITION

### 3.1 OVERVIEW

In this section, we propose a novel learning method for deep sound recognition *BC learning*. Fig. 1 shows the pipeline of BC learning. In standard learning, we select a single training example from the dataset and input it to the model. We then train the model to output 0 or 1. By contrast, in BC learning, we select two training examples from different classes and mix these two examples using a random ratio. We then input the mixed data to the model and train the model to output the mixing ratio. BC learning uses only mixed data and labels, and thus never uses pure data and labels for training. Note that we do not mix any examples in testing phase. First, we provide the details of BC learning in Section 3.2. We mainly explain the method of mixing two sounds, which should be carefully designed to achieve a good performance. Then, in Section 3.3, we explain why BC learning leads to a discriminative feature space.

### 3.2 METHOD DETAILS

#### 3.2.1 MIXING METHOD

BC learning optimizes a model using mini-batch stochastic gradient descent the same way the standard learning does. Each data and label of a mini-batch is generated by mixing two training examples belonging to different classes. Here, we describe how to mix two training examples.

Let $\mathbf{x}_1$ and $\mathbf{x}_2$ be two sounds belonging to different classes randomly selected from the training dataset, and $\mathbf{t}_1$ and $\mathbf{t}_2$ be their one-hot labels. Note that $\mathbf{x}_1$ and $\mathbf{x}_2$ may have already been preprocessed or applied data augmentation, and they have the same length as that of the input of the network. We generate a random ratio $r$ from $U(0, 1)$, and mix two sets of data and labels with this ratio. We mix two labels simply by $r\,\mathbf{t}_1 + (1-r)\,\mathbf{t}_2$, because we aim to train the model to output the mixing ratio. We then explain how to mix $\mathbf{x}_1$ and $\mathbf{x}_2$. The simplest method is $r\,\mathbf{x}_1 + (1-r)\,\mathbf{x}_2$. However, the following mixing formula is slightly better, considering that sound energy is proportional to the square of the amplitude:

$$mix_r(\mathbf{x}_1, \mathbf{x}_2) = \frac{r\,\mathbf{x}_1 + (1-r)\,\mathbf{x}_2}{\sqrt{r^2 + (1-r)^2}}. \tag{1}$$

However, auditory perception of a sound mixed with Eqn. (1) would not be $\mathbf{x}_1 : \mathbf{x}_2 = r : (1-r)$ if the difference of the sound pressure level of $\mathbf{x}_1$ and $\mathbf{x}_2$ is large. For example, if the amplitude of $\mathbf{x}_1$ is 10 times as large as that of $\mathbf{x}_2$ and we mix them with $0.2 : 0.8$, the sound of $\mathbf{x}_1$ would still be dominant in the mixed sound. In this case, training the model with a label of $\{0.2, 0.8\}$ is inappropriate. We then consider using a new coefficient $p(r, G_1, G_2)$ instead of $r$, and mix two sounds by $\frac{p\,\mathbf{x}_1 + (1-p)\,\mathbf{x}_2}{\sqrt{p^2 + (1-p)^2}}$, where $G_1$ and $G_2$ is the sound pressure level of $\mathbf{x}_1$ and $\mathbf{x}_2$ [dB], respectively. We define $p$ so that the auditory perception of the mixed sound becomes $r : (1-r)$. We hypothesize that the ratio of auditory perception for the network is the same as that of amplitude because the main component functions of CNNs, such as conv/fc, relu, max pooling, and average pooling, satisfy homogeneity (i.e., $f(\alpha\,\mathbf{x}) = \alpha f(\mathbf{x})$) if we ignore the bias. We then set up an equation about the ratio of amplitude $p \cdot 10^{\frac{G_1}{20}} : (1-p) \cdot 10^{\frac{G_2}{20}} = r : (1-r)$ using unit conversion from decibels to amplitudes and solve it for $p$. Finally, we obtain the proposed mixing method:

$$mix_r(\mathbf{x}_1, \mathbf{x}_2) = \frac{p\,\mathbf{x}_1 + (1-p)\,\mathbf{x}_2}{\sqrt{p^2 + (1-p)^2}} \quad \text{where} \quad p = \frac{1}{1 + 10^{\frac{G_1 - G_2}{20}} \cdot \frac{1-r}{r}}. \tag{2}$$

We show this mixing method performs better than Eqn. (1) in the experiments.

We calculate the sound pressure level $G_1$ and $G_2$ using A-weighting, considering that human auditory perception is not sensitive to low and high frequency areas. We can also use simpler sound pressure metrics such as root mean square (RMS) energy instead of an A-weighting sound pressure level. However, the performance worsens, as we show in the experiments. We create short windows ($\sim 0.1$ s) on the sound and calculate a time series of A-weighted sound pressure levels $\{g_1, g_2, \ldots, g_t\}$. Then, we define $G$ as the maximum of those time series ($G = max\{g_1, g_2, \ldots, g_t\}$).

### 3.2.2 OPTIMIZATION

We define the $f$ and $\theta$ as the model function and the model parameters, respectively. We input the generated mini-batch data $\{\mathbf{x}^{(i)}\}_{i=1}^n$ to the model and obtain the output $\{f_\theta(\mathbf{x}^{(i)})\}_{i=1}^n$. We expect that our mini-batch ratio labels $\{\mathbf{t}^{(i)}\}_{i=1}^n$ represent the expected class probability distribution. Therefore, we use the KL-divergence between the labels and the model outputs as the loss function, instead of the usual cross-entropy loss. We optimize KL-divergence with back-propagation and stochastic gradient descent because it is differentiable:

$$L = \frac{1}{n}\sum_{i=1}^n D_{\mathrm{KL}}(\mathbf{t}^{(i)}\|f_\theta(\mathbf{x}^{(i)})) = \frac{1}{n}\sum_{i=1}^n \sum_{j=1}^m t_j^{(i)} \log \frac{t_j^{(i)}}{\{f_\theta(\mathbf{x}^{(i)})\}_j}, \tag{3}$$

$$\theta \leftarrow \theta - \eta\frac{\partial L}{\partial \theta}, \tag{4}$$

where $m$ is the number of classes, and $\eta$ is the learning rate.

### 3.3 HOW BC LEARNING WORKS

#### 3.3.1 ENLARGEMENT OF FISHER'S CRITERION

BC leaning leads to an enlargement of Fisher's criterion (*i.e.*, the ratio of the between-class distance to the within-class variance). We explain the reason in Fig. 2. In deep neural networks, linearly-separable features are learned in a hidden layer close to the output layer (An et al., 2015).

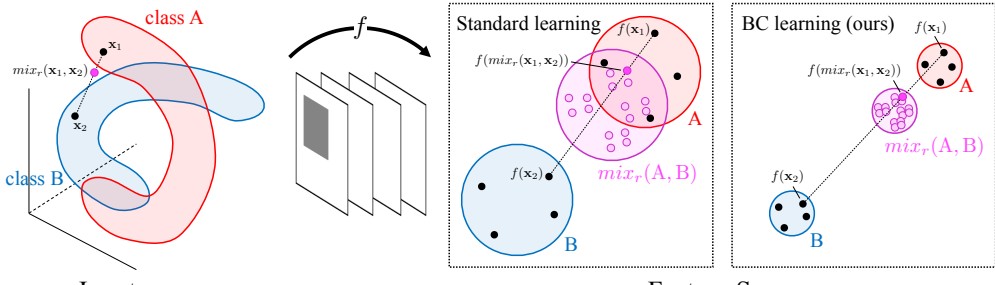

Input space                              Feature Space

Figure 2: BC learning enlarges Fisher's criterion in the feature space, by training the model to output the mixing ratio between two classes. We hypothesize that a mixed sound $mix_r(\mathbf{x}_1, \mathbf{x}_2)$ is projected into the point near the internally dividing point of $f(\mathbf{x}_1)$ and $f(\mathbf{x}_2)$, considering the characteristic of sounds. **Middle**: When Fisher's criterion is small, some mixed examples are projected into one of the classes, and BC learning gives a large penalty. **Right**: When Fisher's criterion is large, most of the mixed examples are projected into between-class points, and BC learning gives a small penalty. Therefore, BC learning leads to such a feature space.

Besides, we can generate a new sound simply by adding the waveform data of two sounds, and humans can recognize both of two sounds and perceive which of two sounds is louder or softer from the mixed sound. Therefore, it is expected that an internally dividing point of the input space almost corresponds to that of the semantic feature space, at least for sounds. Then, the feature distribution of the mixed sounds of class A and class B with a certain ratio would be located near the internally dividing point of the original feature distribution of class A and B, and the variance of the feature distribution of the mixed sounds is proportional to the original feature distribution of class A and B. To investigate whether this hypothesis is correct or not, we visualized the feature distributions of the standard-learned model using PCA. We used the activations of fc6 of EnvNet (Tokozume & Harada, 2017) against training data of ESC-10 (Piczak, 2015b). The results are shown in Fig. 3. The magenta

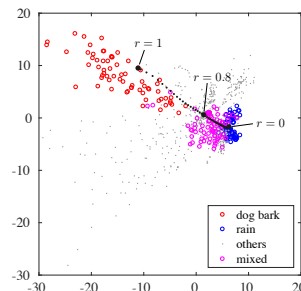

Figure 3: Visualization of the feature space using PCA. The features of the mixed sounds are distributed between two classes.

circles represent the feature distribution of the mixed sounds of *dog bark* and *rain* with a ratio of $0.8 : 0.2$, and the black dotted line represents the trajectory of the feature when we input a mixture of two particular sounds to the model changing the mixing ratio from 0 to 1. This figure shows that the mixture of two sounds is projected into the point near the internally dividing point of two features, and the features of the mixed sounds are distributed between two classes, as we expected.

If Fisher's criterion is small, the feature distribution of the mixed sounds becomes large, and would have a large overlap with one or both of the feature distribution of class A and B (Fig. 2(middle)). In this case, some mixed sounds are projected into one of the classes as shown in this figure, and the model cannot output the mixing ratio. BC learning gives a penalty to this situation because BC learning trains a model to output the mixing ratio. If Fisher's criterion is large, on the other hand, the overlap becomes small (Fig. 2(right)). The model becomes able to output the mixing ratio, and BC learning gives a small penalty. Therefore, BC learning enlarges Fisher's criterion between any two classes in the feature space.

### 3.3.2 REGULARIZATION OF POSITIONAL RELATIONSHIP AMONG FEATURE DISTRIBUTIONS

We expect that BC learning also has the effect of regularizing the positional relationship among the class feature distributions. In standard learning, there is no constraint on the positional relationship among the classes, as long as the features of each two classes are linearly separable. We found that a standard-learned model sometimes misclassifies a mixed sound of class A and class B as a class other than A or B. Fig. 4(lower left) shows an example of transition of output probability of standard-learned model when we input a mixture of two particular training sounds (*dog bark* and *rain*) to the model changing the mixing ratio from 0 to 1. The output probability of *dog bark* monotonically increases and that of *rain* monotonically decreases as we expected, but the model classifies the mixed sound as *baby cry* when the mixing ratio is within the range of $0.45 - 0.8$. This is an undesirable state because there is little possibility that a mixed sound of two classes becomes

a sound of other classes. In this case, we assume that the features of each class are distributed as in Fig. 4(upper left). The decision boundary of class C appears between class A and class B, and the trajectory of the features of the mixed sounds crosses the decision boundary of class C.

BC learning can avoid the situation in which the decision boundary of other class appears between two classes, because BC learning trains a model to output the mixing ratio instead of misclassifying the mixed sound as different classes. We show the transition of the output probability in Fig. 4(lower right), when using the same two examples as that used in Fig. 4(lower left). We assume that the features of each class are distributed as in Fig. 4(upper right). The feature distributions of the three classes make an acute-angled triangle, and the decision boundary of class C does not appear between class A and class B. Note that it is assumed that the di-

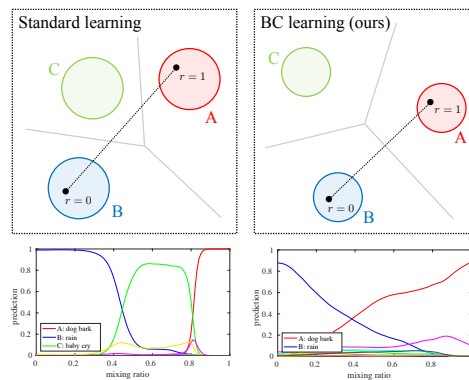

Figure 4: BC learning regularizes the positional relationship of the classes in the feature space, by training the model not to misclassify the mixed sound as different classes. BC learning avoids the situation in which the decision boundary of other class appears between any two classes.

mension of the feature space is greater than or equal to the number of classes minus 1. However, because the network is generally designed as such, it is not a problem. In this way, BC learning enlarges Fisher's criterion, and at the same time, regularizes the positional relationship among the classes in the feature space. Hence, BC learning improves the generalization ability.

## 4 EXPERIMENTS

### 4.1 COMPARISON BETWEEN STANDARD LEARNING AND BC LEARNING

In this section, we train various types of sound recognition networks with both standard and BC learning, and demonstrate the effectiveness of BC learning.

**Datasets.** We used ESC-50, ESC-10 (Piczak, 2015b), and UrbanSound8K (Salamon et al., 2014) to train and evaluate the models. ESC-50, ESC-10, and UrbanSound8K contain a total of $2,000$, $400$, and $8,732$ examples consisting of $50$, $10$, and $10$ classes, respectively. We removed completely silent sections in which the value was equal to $0$ at the beginning or end of examples in the ESC-50 and ESC-10 datasets. We converted all sound files to monaural 16-bit WAV files. We evaluated the performance of the methods using a $K$-fold cross-validation ($K = 5$ for ESC-50 and ESC-10, and $K = 10$ for UrbanSound8K), using the original fold settings. We performed cross-validation $5$ times for ESC-50 and ESC-10, and showed the standard error.

**Preprocessing and data augmentation.** We used a simple preprocessing and data augmentation scheme. Let $T$ be the input length of a network $[s]$. In the training phase, we padded $T/2$ s of zeros on each side of a training sound and randomly cropped a $T$-s section from the padded sound. We mixed two cropped sounds with a random ratio when using BC learning. In the testing phase, we also padded $T/2$ s of zeros on each side of a test sound and cropped $10$ $T$-s sections from the padded sound at regular intervals. We then input these $10$ crops to the network and averaged all softmax outputs. Each input data was regularized into a range of from $-1$ to $+1$ by dividing it by $32,768$, that is, the full range of 16-bit recordings.

**Learning settings.** All models were trained with Nesterov's accelerated gradient using a momentum of $0.9$, weight decay of $0.0005$, and mini-batch size of $64$. The only difference in the learning settings between standard and BC learning is the number of training epochs. BC learning tends to require more training epochs than does standard learning, while standard learning tends to overfit with many training epochs. To validate the comparison, we first identified an appropriate standard learning setting for each network and dataset (details are provided in the appendix), and we doubled the number of training epochs when using BC learning. Later in this section, we examine the relationship between the number of training epochs and the performance.

Table 1: Comparison between standard learning and our BC learning. We performed $K$-fold cross validation using the original fold settings. We performed cross-validation 5 times for the ESC-50 and ESC-10 datasets, and show the standard error. BC learning improves the performance of all models on all datasets, even when we use a strong data augmentation scheme. Our EnvNet-v2 trained with BC learning performs the best and surpasses the human performance on ESC-50.

| Model | Learning | Error rate (%) on | | |
| --- | --- | --- | --- | --- |
| | | ESC-50 | ESC-10 | UrbanSound8K |
| EnvNet (Tokozume & Harada, 2017) | Standard | $29.2 \pm 0.1$ | $12.8 \pm 0.4$ | 33.7 |
| | BC (ours) | $24.1 \pm 0.2$ | $11.3 \pm 0.6$ | 28.9 |
| SoundNet5 (Aytar et al., 2016) | Standard | $33.8 \pm 0.2$ | $16.4 \pm 0.8$ | 33.3 |
| | BC (ours) | $27.4 \pm 0.3$ | $13.9 \pm 0.4$ | 30.2 |
| M18 (Dai et al., 2017) | Standard | $31.5 \pm 0.5$ | $18.2 \pm 0.5$ | 28.8 |
| | BC (ours) | $26.7 \pm 0.1$ | $14.2 \pm 0.9$ | 26.5 |
| Logmel-CNN (Piczak, 2015a) + BN | Standard | $27.6 \pm 0.2$ | $13.2 \pm 0.4$ | 25.3 |
| | BC (ours) | $23.1 \pm 0.3$ | $\mathbf{9.4 \pm 0.4}$ | 23.5 |
| EnvNet-v2 (ours) | Standard | $25.6 \pm 0.3$ | $14.2 \pm 0.8$ | 30.9 |
| | BC (ours) | $\mathbf{18.2 \pm 0.2}$ | $10.6 \pm 0.6$ | $\mathbf{23.4}$ |
| EnvNet-v2 (ours) + strong augment | Standard | $21.2 \pm 0.3$ | $10.9 \pm 0.6$ | 24.9 |
| | BC (ours) | $\mathbf{15.1 \pm 0.2}$ | $\mathbf{8.6 \pm 0.1}$ | $\mathbf{21.7}$ |
| SoundNet8 + Linear SVM (Aytar et al., 2016) | | 25.8 | 7.8 | - |
| Human (Piczak, 2015b) | | 18.7 | 4.3 | - |

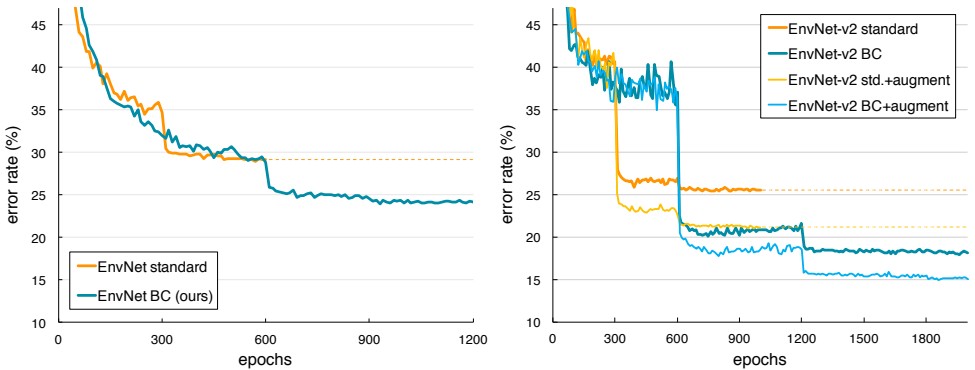

Figure 5: Training curves of EnvNet and EnvNet-v2 on ESC-50 (average of all trials).

### 4.1.1 EXPERIMENT ON EXISTING NETWORKS

First, we trained various types of existing networks. We selected EnvNet (Tokozume & Harada, 2017) as a network using both 1-D and 2-D convolutions, SoundNet5 (Aytar et al., 2016) and M18 (Dai et al., 2017) as networks using only 1-D convolution, and Logmel-CNN (Piczak, 2015a) + BN as a network using log-mel features. Logmel-CNN + BN is an improved version of Logmel-CNN that we designed in which, to convolutional layers, we apply batch normalization (Ioffe & Szegedy, 2015) to the output and remove the dropout (Srivastava et al., 2014). Note that all networks and training codes are our implementation using Chainer v1.24 (Tokui et al., 2015).

The results are summarized in the upper half of Table 1. Our BC learning improved the performance of all networks on all datasets. The performance on ESC-50, ESC-10, and UrbanSound8K was improved by 4.5–6.4%, 1.5–4.0%, and 1.8–4.8%, respectively. We show the training curves of EnvNet on ESC-50 in Fig. 5(left). Note that the curves show the average of all trials.

### 4.1.2 EXPERIMENT ON A DEEPER NETWORK

To investigate the effectiveness of BC learning on deeper networks, we constructed a deep sound recognition network based on EnvNet, which we refer to as *EnvNet-v2*, and trained it with both

standard and BC learning. The main differences between EnvNet and EnvNet-v2 are as follows: 1) EnvNet uses a sampling rate of 16 kHz for the input waveforms, whereas EnvNet-v2 uses 44.1 kHz; and 2) EnvNet consists of 7 layers, whereas EnvNet-v2 consists of 13 layers. A detailed configuration is provided in the appendix.

The results are also shown in the upper half of Table 1, and the training curves on ESC-50 are given in Fig. 5(right). The performance was also improved with BC learning, and the degree of the improvement was greater than other networks (7.4%, 3.6%, and 7.5% on ESC-50, ESC-10, and UrbanSound8K, respectively). The error rate of EnvNet-v2 trained with BC learning was the lowest on ESC-50 and UrbanSound8K among all the models including Logmel-CNN + BN, which uses powerful hand-crafted features. Moreover, the error rate on ESC-50 (18.2%) is comparable to human performance reported by Piczak (2015b) (18.7%). The point is not that our EnvNet-v2 is well designed, but that our BC learning successfully elicits the true value of a deep network.

### 4.1.3 EXPERIMENT WITH STRONG DATA AUGMENTATION

We compared the performances of standard and BC learning when using a stronger data augmentation scheme. In addition to zero padding and random cropping, we used scale augmentation with a factor randomly selected from $[0.8, 1.25]$ and gain augmentation with a factor randomly selected from $[-6 \text{ dB}, +6 \text{ dB}]$. Scale augmentation was performed before zero padding (thus, before mixing when employing BC learning) using linear interpolation, and gain augmentation was performed just before inputting to the network (thus, after mixing when using BC learning).

The results for EnvNet-v2 are shown in the lower half of Table 1, and the training curves on ESC-50 are given in Fig. 5(right). With BC learning, the performance was significantly improved even when we used a strong data augmentation scheme. Furthermore, the performance on ESC-50 (15.1%) surpasses the human performance (18.7%). BC learning performs well on various networks, datasets, and data augmentation schemes, and using BC learning is always beneficial.

### 4.1.4 RELATIONSHIP WITH # OF TRAINING EPOCHS

We investigated the relationship between the performance and the number of training epochs, because the previously described experiments were conducted using different numbers of training epochs (we used $2\times$ training epochs for BC learning). Fig. 6 shows the error rate of EnvNet on ESC-10 and ESC-50 with various numbers of training epochs. This figure shows that for standard learning, approximately 600 training epochs are sufficient for both ESC-

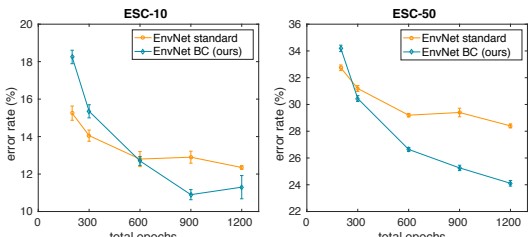

Figure 6: Error rate *vs.* # of training epochs.

10 and ESC-50. However, this number is insufficient for BC learning. Although BC learning performed better than standard learning with 600 epochs, improved performance was achieved when using more training epochs (900 and 1,200 epochs for ESC-10 and ESC-50, respectively). However, if the number of training epochs was small, the performance of BC learning was lower than that of standard learning. We can say that BC learning always improves the performance as long as we use a sufficiently large number of training epochs. Additionally, the number of training epochs needed would become large when there are many classes.

### 4.2 ABLATION ANALYSIS

To understand the part that is important for BC learning, we conducted an ablation analysis. We trained EnvNet on ESC-50 using various settings. All results are shown in Table 2. We also performed 5-fold cross-validation five times and show the standard error.

**Mixing method.** We compared the mixing formula (Eqn. 1 *vs.* Eqn. 2, which consider the sound pressure levels of two sounds) and the calculation method for sound pressure levels (RMS *vs.* A-weighting). As shown in Table 2, the proposed mixing method using Eqn. 2 and A-weighting performed the best. Considering the difference in the sound pressure levels is important for BC learning, and the method used to define the sound pressure levels also has an effect on the performance.

**Label.** We compared the different labels that we applied to the mixed sound. As shown in Table 2, the proposed ratio label of $\mathbf{t} = r\,\mathbf{t}_1 + (1 - r)\,\mathbf{t}_2$ performed the best. When we applied a single label of the dominant sound (*i.e.*, $\mathbf{t} = \mathbf{t}_1$ if $r > 0.5$, otherwise $\mathbf{t} = \mathbf{t}_2$) and trained the model using softmax cross entropy loss, the performance was improved compared to that of standard learning. When we applied a multi-label (*i.e.*, $\mathbf{t} = \mathbf{t}_1 + \mathbf{t}_2$) and trained the model using sigmoid cross entropy loss, the performance was better than when using a single label. However, the performance was worse than when using our ratio label in both cases. The model can learn the between-class examples more efficiently when using our ratio label.

**Number of mixed classes.** We investigated the relationship between the performance and the number of sound classes that we mixed. $N = 1$ in Table 2 means that we mixed two sounds belonging to the same class, which is similar to Takahashi et al. (2016). $N = 1$ or $2$ means that we completely randomly selected two sounds to be mixed; sometimes these two sounds were the same class. $N = 2$ or $3$ means that we mixed two and three sounds belonging to different classes with probabilities of $0.5$ and $0.5$, respectively. When we mixed three sounds, we generated a mixing ratio from Dir(1,1,1) and mixed three sounds using a method that is an extended version of Eqn. 2 to three classes. As shown in Table 2, the proposed $N = 2$ performed the best. $N = 2$ or $3$ also achieved a good performance. It is interesting to note that the performance of $N = 3$ is worse than that of $N = 2$ despite the larger variation in training data. We believe that the most important factor is not the training data variation but rather the enlargement of Fisher's criterion and the regularization of the positional relationship among the feature distributions. Mixing more than two sounds leads to increased training data variation, but we expect that cannot efficiently achieve them.

**Where to mix.** Finally, we investigated what occurs when we mix two examples within the network. We input two sounds to be mixed into the model and performed the forward calculation to the mixing point. We then mixed the activations of two sounds at the mixing point and performed the rest of the forward calculation. We mixed two activations $\mathbf{h}_1$ and $\mathbf{h}_2$ simply by $r\,\mathbf{h}_1 + (1-r)\,\mathbf{h}_2$. As shown in Table 2, the performance tended to improve when we mixed two examples at the layer near the input layer. The performance was the best when we mixed in the input space. Mixing in the input space is the best choice, not only because it performs the best, but also because it does not require additional forward/backward computation and is easy to implement.

Table 2: Ablation analysis. We trained EnvNet on ESC-50 using various settings. The results show that the training data variation is not the only matter.

| Comparison of | Setting | Err. rate (%) |
|---|---|---|
| Mixing method | Eqn. (1) | $26.8 \pm 0.1$ |
| | (2) + RMS | $26.5 \pm 0.2$ |
| | (2) + A-weighting (proposed) | $\mathbf{24.1 \pm 0.2}$ |
| Label | Single | $26.5 \pm 0.2$ |
| | Multi | $25.0 \pm 0.3$ |
| | Ratio (proposed) | $\mathbf{24.1 \pm 0.2}$ |
| # mixed classes | $N = 1$ | $27.3 \pm 0.2$ |
| | $N = 1$ or $2$ | $24.8 \pm 0.3$ |
| | $N = 2$ (proposed) | $\mathbf{24.1 \pm 0.2}$ |
| | $N = 2$ or $3$ | $\mathbf{24.1 \pm 0.2}$ |
| | $N = 3$ | $25.3 \pm 0.2$ |
| Where to mix | Input (proposed) | $\mathbf{24.1 \pm 0.2}$ |
| | pool2 | $27.1 \pm 0.3$ |
| | pool3 | $28.7 \pm 0.3$ |
| | pool4 | $28.8 \pm 0.2$ |
| | fc5 | $28.5 \pm 0.1$ |
| | fc6 | $28.6 \pm 0.2$ |
| Standard learning | | $29.2 \pm 0.1$ |

## 5 CONCLUSION

We proposed a novel learning method for deep sound recognition, called BC learning. Our method improved the performance on various networks, datasets, and data augmentation schemes. Moreover, we achieved a performance surpasses the human level by constructing a deeper network named EnvNet-v2 and training it with BC learning. BC learning is a simple and powerful method that improves various sound recognition methods and elicits the true value of large-scale networks. Furthermore, BC learning is innovative in that a discriminative feature space can be learned from between-class examples, without inputting pure examples. We assume that the core idea of BC learning is generic and could contribute to the improvement of the performance of tasks of other modalities.

ACKNOWLEDGEMENT

This work was supported by JST CREST Grant Number JPMJCR1403, Japan.

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

## A    LEARNING SETTINGS

Table 3 shows the detailed learning settings of standard learning. We trained the model by beginning with a learning rate of *Initial LR*, and then divided the learning rate by 10 at the epoch listed in *LR schedule*. To improve convergence, we used a $0.1\times$ smaller learning rate for the first *Warmup* epochs. We then terminated training after *# of epochs* epochs. We doubled *# of epochs* and *LR schedule* when using BC learning, as we mentioned in the paper.

Table 3: Details of learning settings.

| Dataset | Model | # of epochs | Initial LR | LR schedule | Warmup |
|---------|-------|-------------|------------|-------------|--------|
| ESC-50 | EnvNet | 600 | 0.01 | {300, 450} | 0 |
| | SoundNet5 | 300 | 0.1 | {150, 225} | 0 |
| | M18 | 400 | 0.1 | {200, 300} | 0 |
| | Logmel-CNN | 300 | 0.01 | {150, 225} | 0 |
| | EnvNet-v2 | 1,000 | 0.1 | {300, 600, 900} | 10 |
| ESC-10 | EnvNet | 600 | 0.01 | {300, 450} | 0 |
| | SoundNet5 | 300 | 0.1 | {150, 225} | 0 |
| | M18 | 400 | 0.1 | {200, 300} | 0 |
| | Logmel-CNN | 300 | 0.01 | {150, 225} | 0 |
| | EnvNet-v2 | 600 | 0.01 | {300, 450} | 0 |
| UrbanSound8K | EnvNet | 400 | 0.01 | {200, 300} | 0 |
| | SoundNet5 | 200 | 0.1 | {100, 150} | 0 |
| | M18 | 300 | 0.1 | {150, 225} | 0 |
| | Logmel-CNN | 200 | 0.01 | {100, 150} | 0 |
| | EnvNet-v2 | 600 | 0.1 | {180, 360, 540} | 10 |

## B    CONFIGURATION OF ENVNET-V2

Table 4 shows the configuration of our EnvNet-v2 used in the experiments. EnvNet-v2 consists of 10 convolutional layers, 3 fully connected layers, and 5 max-pooling layers. We use a sampling rate of 44.1 kHz, which is the standard recording setting, and a higher resolution than existing networks (Piczak, 2015a; Aytar et al., 2016; Dai et al., 2017; Tokozume & Harada, 2017), in order to use rich high-frequency information. The basic idea is motivated by EnvNet (Tokozume & Harada, 2017), but the advantages of other successful networks are incorporated. First, we extract short-time frequency features with the first two temporal convolutional layers and a pooling layer (conv1–pool2). Second, we swap the axes and convolve in time and frequency domains with the later layers (conv3–pool10). In this part, we hierarchically extract the temporal features by stacking the convolutional and pooling layers with decreasing their kernel size in a similar manner to SoundNet (Aytar et al., 2016). Furthermore, we stack multiple convolutional layers with a small kernel size in a similar manner to M18 (Dai et al., 2017) and VGG (Simonyan & Zisserman, 2015), to extract more rich features. Finally, we produce output predictions with fc11–fc13 and the following softmax activation. Single output prediction is calculated from 66,650 input samples (approximately 1.5 s at 44.1 kHz). We do not use padding in convolutional layers. We apply ReLU activation for all the hidden layers and batch normalization (Ioffe & Szegedy, 2015) to the output of conv1–conv10. We also apply 0.5 of dropout (Srivastava et al., 2014) to the output of fc11 and fc12. We use a weight initialization of He et al. (2015) for all convolutional layers. We initialize the weights of each fully connected layer using Gaussian distribution with a standard deviation of $\sqrt{1/n}$, where $n$ is the input dimension of the layer.

Table 4: Configuration of EnvNet-v2. Data shape represents the dimension in (channel, frequency, time).

| Layer | ksize | stride | # of filters | Data shape |
|---|---|---|---|---|
| Input | | | | (1, 1, 66,650) |
| conv1 | (1, 64) | (1, 2) | 32 | |
| conv2 | (1, 16) | (1, 2) | 64 | |
| pool2 | (1, 64) | (1, 64) | | (64, 1, 260) |
| swapaxes | | | | (1, 64, 260) |
| conv3, 4 | (8, 8) | (1, 1) | 32 | |
| pool4 | (5, 3) | (5, 3) | | (32, 10, 82) |
| conv5, 6 | (1, 4) | (1, 1) | 64 | |
| pool6 | (1, 2) | (1, 2) | | (64, 10, 38) |
| conv7, 8 | (1, 2) | (1, 1) | 128 | |
| pool8 | (1, 2) | (1, 2) | | (128, 10, 18) |
| conv9, 10 | (1, 2) | (1, 1) | 256 | |
| pool10 | (1, 2) | (1, 2) | | (256, 10, 8) |
| fc11 | - | - | 4096 | (4,096, ) |
| fc12 | - | - | 4096 | (4,096, ) |
| fc13 | - | - | # of classes | (# of classes, ) |

