# OpenReview forum: "Learning from Between-class Examples for Deep Sound Recognition"
_ICLR.cc/2018/Conference — Accept (Poster)_

### Official Review · AnonReviewer1 · 2017-11-25
**Novel Approach to Using Limited Training Data By Changing Task for Better Class Discrimination**

**Rating:** 9
**Confidence:** 4

**Review:**

Overall: Authors defined a new learning task that requires a DNN to predict mixing ratio between sounds from two different classes. Previous approaches to training data mixing are (1) from random classes, or (2) from the same class. The presented approach mixes sounds from specific pairs of classes to increase discriminative power of the final learned network. Results look like significant improvements over standard learning setups.

Detailed Evaluation: The approach presented is simple, clearly presented, and looks effective on benchmarks. In terms of originality, it is different from warping training example for the same task and it is a good extension of previously suggested example mixing procedures with a targeted benefit for improved discriminative power. The authors have also provided extensive analysis from the point of views (1) network architecture, (2) mixing method, (3) number of labels / classes in mix, (4) mixing layers -- really well done due-diligence across different model and task parameters.

Minor Asks:
(1) Clarification on how the error rates are defined. Especially since the standard learning task could be 0-1 loss and this new BC learning task could be based on distribution divergence (if we're not using argmax as class label).
(2) #class_pairs targets as analysis - The number of epochs needed is naturally going to be higher since the BC-DNN has to train to predict mixing ratios between pairs of classes. Since pairs of classes could be huge if the total number of classes is large, it'll be nice to see how this scales. I.e. are we talking about a space of 10 total classes or 10000 total classes? How does num required epochs get impacted as we increase this class space?
(3) Clarify how G_1/20 and G_2/20 is important / derived - I assume it's unit conversion from decibels.
(4) Please explain why it is important to use the smoothed average of 10 softmax predictions in evaluation... what happens if you just randomly pick one of the 10 crops for prediction?

---

> ### Author Response · Authors · 2017-12-17
> **Response to AnonReviewer1**
>
> Thanks for your positive review. Our method is novel in that we train the model to output the mixing ratio between two different classes.
>
> Answers for minor asks:
> (1) We do not define error rate of BC learning in training phase. In testing phase, the error rate definition of BC learning is the same as that of standard learning because we do not mix any sounds in testing phase.
>
> (2) Thanks for helpful advice. Although we could not try more than 50 classes, we have investigated the relationship between performance and the number of training epochs not only on ESC-50 (50 classes, Fig. 6) but also ESC-10 (10 classes). As a result, the sufficient number of training epochs for BC learning on ESC-10 was 900, which is smaller than that for BC learning on ESC-50 (1,200 epochs), whereas that for standard learning was 600 epochs on both ESC-50 and ESC-10. We assume that the number of training epochs needed would become large when there are many classes, as you have suggested. We will add this discussion to the final version.
>
> (3) Yes, G_1 and G_2 are derived from unit conversion from decibels to amplitudes. We will clarify it. Please see also the reply to AnonReviewer2.
>
> (4) We have tried random 1-crop testing and center 1-crop testing on EnvNet on ESC-50 (standard learning). The error rates of random 1-crop testing and center 1-crop testing were 41.3% and 39.2%, respectively, whereas that of 10-crop testing was 29.2% as in the paper. Averaging the predictions of multiple windows leads to a stable performance. We assume this is because we cannot know where the target sound exists in a testing sound, and the target sound sometimes has a long duration.

---

### Official Review · AnonReviewer3 · 2017-11-27
**Interesting data augmentation technique, but lacks of deep insights on how and why does it work.**

**Rating:** 4
**Confidence:** 4

**Review:**

This manuscript proposes a method to improve the performance of a generic learning method by generating "in between class" (BC) training samples. The manuscript motivates the necessity of such technique and presents the basic intuition. The authors show how the so-called BC learning helps training different deep architectures for the sound recognition task.

My first remark regards the presentation of the technique. The authors argue that it is not a data augmentation technique, but rather a learning method. I strongly disagree with this statement, not only because the technique deals exactly with augmenting data, but also because it can be used in combination to any learning method (including non-deep learning methodologies). Naturally, the literature review deals with data augmentation technique, which supports my point of view.

In this regard, I would have expected comparison with other state-of-the-art data augmentation techniques. The usefulness of the BC technique is proven to a certain extent (see paragraph below) but there is not comparison with state-of-the-art. In other words, the authors do not compare the proposed method with other methods doing data augmentation. This is crucial to understand the advantages of the BC technique.

There is a more fundamental question for which I was not able to find an explicit answer in the manuscript. Intuitively, the diagram shown in Figure 4 works well for 3 classes in dimension 2. If we add another class, no matter how do we define the borders, there will be one pair of classes for which the transition from one to another will pass through the region of a third class. The situation worsens with more classes. However, this can be solved by adding one dimension, 4 classes and 3 dimensions seems something feasible. One can easily understand that if there is one more class than the number of dimensions, the assumption should be feasible, but beyond it starts to get problematic. This discussion does not appear at all in the manuscript and it would be an important limitation of the method, specially when dealing with large-scale data sets.

Overall I believe the paper is not mature enough for publication.

Some minor comments:
- 2.1: We introduce --> We discussion
- Pieczak 2015a did not propose the extraction of MFCC.
- the x_i and t_i of section 3.2.2 should not be denoted with the same letters as in 3.2.1.
- The correspondence with a semantic feature space is too pretentious, specially since no experiment in this direction is shown.
- I understand that there is no mixing in the test phase, perhaps it would be useful to recall it.

---

> ### Author Response · Authors · 2017-12-17
> **Response to AnonReviewer3**
>
> Thanks for your helpful review.
>
> - Regarding the presentation of BC learning:
> It is true that BC learning is a data augmentation method as you have suggested, from a view point of using augmented data. However, our method is novel in that we change the objective of training by training the model to output the mixing ratio, which is fundamentally a different idea from previous data augmentation methods. The novelty or key point of our method is not mixing multiple sounds, but rather learning method of training the model to output the mixing ratio. That is why we represent our method as "learning method." We intuitively describe why such a learning method is effective in Section 3.3 and demonstrate the effectiveness of BC learning through wide-ranging experiments.
>
> - Regarding comparison with other data augmentation methods:
> First, we compared BC learning with other data augmentation methods that mix multiple sounds in ablation analysis (see Table 2), and showed that our method of mixing just two classes with equation 2 and training the model to output the mixing ratio performs the best.
>
> Second, our BC learning can be combined with any data augmentation methods that do not mix multiple sounds by mixing two augmented data. In Section 4.1.3, we demonstrated that BC learning is even "compatible" with a strong data augmentation, which we believe is more important than being "stronger" than that. This data augmentation method uses scale and amplitude augmentation similar to Salamon & Bello (2017) in addition to padding and cropping, and thus, it is close to the state-of-the-art level. As shown in Table 1, the error rates of DSRNet when using only BC learning (18.2%, 10.6%, and 23.4% on ESC-50, ESC-10, and UrbanSound8K, respectively) were lower than those when using the strong data augmentation (21.2%, 10.9%, and 24.9%). Furthermore, as a result of combination of BC learning and the strong data augmentation, we achieved a further higher performance (15.1%, 8.6%, and 21.7%). In this way, we demonstrated the strongness and compatibility of BC learning with other data augmentation techniques through various experiments.
>
> Here, we assume that the effect of BC learning is even strengthened when using a stronger data augmentation scheme. Because the potential within-class variance becomes large when using a strong data augmentation, the overlap between the feature distribution of each class and that of mixed sounds tends to become large and it becomes more difficult for model to output the mixing ratio (see also Fig. 2). Therefore, the effect of enlargement of Fisher’s criterion would become stronger.
>
> - Regarding the limitation of BC learning:
> Thanks for your advice. What you have pointed out is correct. However, the dimension of the feature space d is generally designed to be larger than the number of classes c (e.g., EnvNet/DSRNet: 4096; SoundNet: 256; M18: 512; and Logmel-CNN: 5000). If d < c-1, the features cannot sufficiently represent categorical information, and the model would not be able to achieve a good performance. We have tried to train an EnvNet whose dimension of fully connected layer was made less than 49 on ESC-50 with standard learning, but the loss did not begin to decrease. It is not a matter of BC learning. Furthermore, even if there is a network whose d is smaller than c-1, BC learning would enlarge Fisher's criterion and regularize the positional relationship as much as possible. Therefore, we do not think it is an important limitation of BC learning.
>
>
> Thanks for other helpful comments. We will reflect them to the final version. Note than we showed the correspondence with a semantic feature space by visualizing the features of mixed sounds in Fig. 3.

---

> > ### Comment · AnonReviewer3 · 2018-01-15
> > **More clear now**
> >
> > After reading all the reviews and the authors' responses I have a more clear view of the paper now. I changed my mind, I see now the interest of having this paper accepted.
> >
> > Having said that, I strongly encourage the authors to modify the text so that two key aspects of the method are clearly explained.
> > First, as you said in your comments: The novelty or key point of our method is not mixing multiple sounds, but rather learning method of training the model to output the mixing ratio.
> > Second, the limitation (which now is not a problem, but could be for some other applications/architectures): the dimension of the feature space d is generally designed to be larger than the number of classes c.

---

> > > ### Author Response · Authors · 2018-01-18
> > > **Response**
> > >
> > > Thanks for your positive comments and advice. We will modify the two points you have suggested (will be uploaded in a few days).

---

### Official Review · AnonReviewer2 · 2017-11-27
**Learning from Between-class Examples increases the Fisher’s criterion. BC learning regularizes the positional relationship of the classes in the feature space, by training the model not to misclassify the mixed sound as different classes. The presentation and discussion on the proposed method is good (even if the organisation of the paper could be improved). Some simplifications can be conducted (in the mixture). The main idea is good and novel, and relevant for ICLR.**

**Rating:** 8
**Confidence:** 4

**Review:**

The propose data augmentation and BC learning is relevant, much robust than frequency jitter or simple data augmentation.

In equation 2, please check the measure of the mixture. Why not simply use a dB criteria ?

The comments about applying a CNN to local features or novel approach to increase sound recognition could be completed with some ICLR 2017 work towards injected priors using Chirplet Transform.

The authors might discuss more how to extend their model to image recognition, or at least of other modalities as suggested.

Section 3.2.2 shall be placed later on, and clarified.

Discussion on mixing more than two sounds leads could be completed by associative properties, we think... ?

---

> ### Author Response · Authors · 2017-12-17
> **Response to AnonReviewer2**
>
> Thanks for your positive review.
>
> - Regarding equation 2:
> We use 10^(G_1/20) and 10^(G_2/20) instead of simple G_1 and G_2 to convert decibels to amplitudes. We hypothesize that the ratio of auditory perception for the network is the same as the ratio of amplitude, and define p so that the auditory perception of the mixed sound becomes r: (1-r). This is because the main component functions of CNNs, such as conv/fc, relu, max pooling, and average pooling, satisfy homogeneity (i.e., f(ax) = af(x)) if we ignore the bias. We will clarify the derivation and meaning of equation 2.
>
> - Regarding how to extend BC learning to other modalities:
> We assume that BC learning can also be applied to image classification. Image data can be treated as 2-D waveforms along x- and y- axis that contain various areas of frequency information in quite a similar manner to sound data. In addition, recent studies on speech/sound recognition have demonstrated that each filter of CNNs learns to respond to a particular frequency area (e.g., Sainath et al., 2015b). Considering them, we assume that CNNs have aspect of recognizing images by treating them as waveforms in a similar manner to how they recognize sounds, and what works on sounds must also work on images. A simple mixing method (r x_1 + (1-r) x_2) would work well, but we assume that a mixing method that treats the images as waveforms (similar to equation 2) leads to a further performance improvement.
>
> - Regarding mixing more than two classes:
> Mixing more than two classes would have a similar effect to mixing just two classes. However, the number of class combinations dramatically increases, and it would become difficult to train. Mixing just two classes can directory impose a constraint on the feature distribution (as we describe in Section 3.3). Therefore, we assume that mixing just two classes is the most efficient. Experimental results also show that mixing two classes performs better than mixing three classes (see Table 2).
>
> Thanks for other helpful comments. We will reflect them to the final version.

---

### Author Response · Authors · 2018-01-05
**Revised paper uploaded**

We have uploaded a revised version of the paper.

Major changes:
- Section 2.1: modified the description of Piczak's work.
- Section 3.1: noted that there is no mixing in testing phase.
- Section 3.2.1: clarified the deviation and meaning of equation 2.
- Section 3.2.2: modified the indices of x and t.
- Section 4.1.4: added experiments and discussion on # of training epochs vs. # of classes.
- Total: 10 pages -> 9 pages (for main part)

---

### Public Comment · (anonymous) · 2018-01-20
**iclr 2018 reproducibility challenge**

Hello,
I am trying to reproduce your results, as part of the reproducibility challenge.
The database that I am using is ESC-50, with DSRNet.
I have several questions regarding the training/results of the baseline network (no BC learning):
1. Did you split the data to 5 cross validation sections, each one with 400 samples (2000/5)? Did you have test samples, besides the validation folds? or the testing was done only on the validation folds?
2. Did the epochs in Figure 5 represent single iteration over the data base, or 5 iterations, because of the cross validation?
3. How did you calculate the Error rate in table 1 and figure 5? Is it the error of the current fold? average between the errors of the 5 folds?
4. When you write that during testing, you cropped 10 T-s sections, Did you do that on the validation fold every epoch, or on a different testing data?
Thanks

---

> ### Author Response · Authors · 2018-01-21
> **Re: iclr 2018 reproducibility challenge**
>
> Thanks for your questions.
> 1. Each fold has 400 samples. We did not have test samples, and testing was done only on the validation fold. The model was trained with 4 folds (1600 samples) and tested with 1 fold (400 samples). We used the original fold settings defined by the proposer of ESC-50 (not random division).
> 2. Epochs in Figure 5 represent single iteration over the 1600 training samples.
> 3. We performed 5-fold cross-validation 5 times. Thus, the errors in Table 1 and Figure 5 represent the average of (5x5=) 25 errors.
> 4. We did 10-crop testing every epoch.

---

### Author Response · Authors · 2018-01-22
**Revised paper uploaded**

We have revised the paper, considering the comments from AnonReviewer3.
Major changes:
- The last paragraph of Section 1: modified the description about the novelty of our paper.
- Section 3.3.2: added the description about the dimension of the feature space.

---

### Public Comment · ~Daisuke_Niizumi1 · 2018-03-16
**Similar concept with "mixup: Beyond Empirical Risk Minimization"**

Hi, for me your work seems to be similar to this paper:
Hongyi Zhang, Moustapha Cisse, Yann N. Dauphin, David Lopez-Paz, mixup: Beyond Empirical Risk Minimization, 2017, https://arxiv.org/abs/1710.09412
This mixes two examples regardless of class from training set.
I guess it would be better if you could also mention regarding relationship with this.

---

> ### Author Response · Authors · 2018-03-17
> **Re: Similar concept with "mixup: Beyond Empirical Risk Minimization"**
>
> Thank you for pointing it out. Our BC learning is actually similar to mixup, but our work is different from mixup in two points:
> - We intuitively described why BC learning works well from a viewpoint of feature distributions.
> - We carefully designed how to mix two training examples. We showed that our method is better than the simplest method of r x1 + (1-r) x2.
> Unfortunately, we cannot mention the relationship with mixup in the conference paper because the camera-ready submission has already been closed.

---

### Decision · Program_Chairs · 2018-01-29
**ICLR 2018 Conference Acceptance Decision**

**Decision:**

Accept (Poster)

**Comment:**

meta score: 8

This is a good paper which augments the data by mixing sound classes, and then learns the  mixing ratio.  Experiments performed on a number of sound classification results

Pros
 - novel approach, clearly explained
 - very good set of experimentation with excellent results
 - good approach to mixing using perceptual criteria

Cons
 - discussion doesn't really generalise beyond sound recognition